# ARE LLMs REALLY NOT KNOWLEDGEABLE? MINING THE SUBMERGED KNOWLEDGE IN LLMs' MEMORY

**Xingjian Tao[1]    Yiwei Wang[3]    Yujun Cai[4]    Zhicheng Yang[1]    Jing Tang[1,2]\***

[1]The Hong Kong University of Science and Technology (Guangzhou)
[2]The Hong Kong University of Science and Technology    [3]University of California, Merced
[4]The University of Queensland
`taoxj2001@outlook.com, wangyw.evan@gmail.com, jingtang@ust.hk`
[https://github.com/taoxj2001/Hits_at_k](https://github.com/taoxj2001/Hits_at_k)

## ABSTRACT

Large language models (LLMs) have shown promise as parametric knowledge bases, but often underperform on question answering (QA) tasks due to hallucinations and uncertainty. While prior work attributes these failures to knowledge gaps in the model's parameters, we uncover a complementary phenomenon: LLMs frequently retain correct knowledge even when generating incorrect or "unsure" answers. By analyzing the token-level output distributions, we find that correct answers often appear among high-probability candidates, despite not being selected. Motivated by this, we propose Hits@$k$, a novel metric to evaluate latent knowledge retention independent of answer surface form. Our experiments reveal that LLMs possess significantly more factual knowledge than is reflected by standard QA accuracy. Building on these insights, we further examine the prevailing few-shot QA paradigm. We find that prompting strategies which allow "unsure" outputs can inadvertently suppress correct answers by discouraging low-confidence generation. We design a set of quantitative experiments to measure this suppression effect, offering practical guidance for future prompt and decoding design in knowledge-intensive tasks.

## 1 INTRODUCTION

Large language models (LLMs; Touvron et al. 2023a; Chiang et al. 2023; Almazrouei et al. 2023; MosaicML 2023; Touvron et al. 2023b; OpenAI 2022; Google 2023) have emerged as potential alternatives to traditional knowledge bases, demonstrating capabilities in encoding and retrieving vast amounts of factual information through their parameters. The ability to accurately access and utilize this knowledge is crucial for reliable deployment of LLMs in real-world applications, from question answering to decision support systems. However, these models frequently produce incorrect answers or hallucinations in knowledge-intensive tasks, severely limiting their practical utility. Recent studies have explored multiple approaches to enhance LLMs' knowledge utilization, including domain-specific fine-tuning Kumar et al. (2024), prompt engineering strategies Zhang et al. (2023a), and architectural modifications Zhong et al. (2023). These methods operate under the assumption that answer inaccuracies stem from insufficient knowledge in model parameters, leading to solutions focused on expanding model capacity or training data.

Our systematic investigation reveals fundamental limitations in this understanding of LLMs' knowledge utilization. Analysis of model outputs demonstrates that even when generating incorrect answers, LLMs often maintain access to accurate information within their probability distributions over vocabulary tokens. In state capital queries, for instance, while models might output "Seattle" as Washington's capital, they consistently assign high probability scores to the correct answer "Olympia". This pattern persists across various knowledge domains and model architectures, indicating a systematic gap between knowledge storage and expression rather than simple knowledge absence.

---

\* Corresponding Author: Jing Tang.

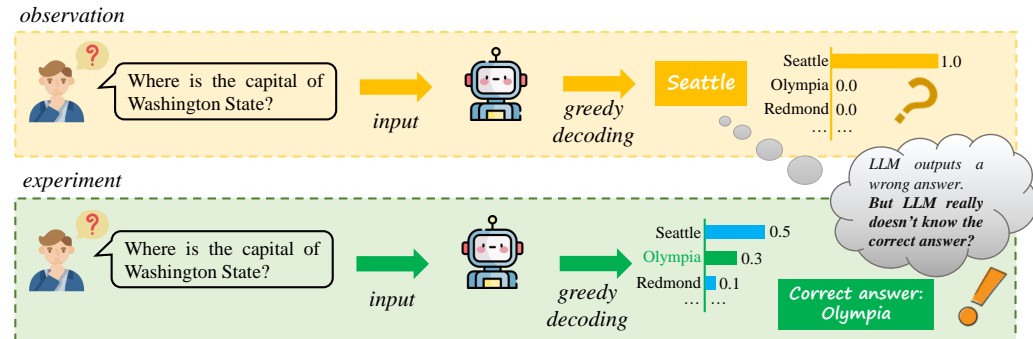

Figure 1: An example illustrating a scenario where a model possesses potentially correct memories yet fails to provide the correct answer.

To quantify this phenomenon, we propose Hits@$k$ to evaluate knowledge retention independent of answer accuracy. Extensive experiments across multiple datasets demonstrate the prevalence of this storage-expression gap. On DBpedia, LLAMA3-8B achieves only 17.2% standard accuracy (Hits@1) but reaches 57.9% for Hits@5, revealing substantially more stored knowledge than conventional metrics suggest. This disparity is particularly pronounced in domain-specific tasks and varies systematically with data popularity, offering insights into how LLMs organize and access their stored knowledge. Traditional evaluation methods, focusing solely on final outputs, significantly underestimate the knowledge actually encoded in model parameters.

Building on these insights, we further examine the widely adopted few-shot QA paradigm. We observe that prompting strategies which permit "unsure" responses may inadvertently suppress the generation of low-confidence answers, thereby inhibiting the expression of correct knowledge. In particular, we find that when the model outputs "unsure", tokens ranked highly in the logit distribution—though not selected as the top-1 prediction—often contain the correct answer. To quantify this effect, we design a set of experiments where "unsure"-related tokens are filtered out during decoding. Under this strategy, a subset of previously masked correct answers can be successfully recovered. These findings highlight a potential trade-off between cautious generation and knowledge expressiveness, and provide actionable guidance for future prompt design and decoding strategies in knowledge-intensive tasks.

This work makes the following key contributions:

- We identify and analyze a systematic gap between knowledge storage and expression in large language models (LLMs).
- We propose Hits@$k$, a novel metric for quantifying latent knowledge retention independent of output accuracy.
- We conduct a comprehensive analysis of the factors that influence the alignment between stored knowledge and generated answers.
- We quantitatively demonstrate that existing prompting paradigms can inadvertently suppress correct answers by enabling "unsure" responses, revealing a memory-masking effect.

## 2 EXPLORING MEMORY IN LLMS

### 2.1 KNOWLEDGE STORAGE AND EXPRESSION

Recent studies have explored using LLMs as knowledge bases, highlighting their potential to encode information within parameters through pre-training Petroni et al. (2019); Wang et al. (2020). While these models demonstrate impressive capabilities in question answering tasks, they often struggle with consistency and hallucination. Prior work frequently attributes such failures to knowledge gaps in the model's parameters Sun et al. (2023); Li et al. (2024), suggesting that expanding model capacity or training data could address these issues.

Our investigation reveals that model failures may stem from expression issues rather than knowledge gaps. Through systematic analysis of model outputs, we find that LLMs often retain correct information in their parameters even when generating incorrect answers. As shown in Figure 1, when asked about Washington state's capital, while the model outputs "Seattle", it assigns a high probability score to the correct answer "Olympia". Such cases indicate the need for a deeper understanding of how knowledge is stored and expressed in these models.

## 2.2 ANALYZING MODEL'S INTERNAL KNOWLEDGE

We investigate this phenomenon by examining the logits, which represent token probabilities produced during the model's answer generation process. In LLMs, these logits reflect the model's internal knowledge state before the final output selection. Our analysis of these distributions reveals a consistent pattern. Even when the model fails to output the correct answer, it often assigns significant probability scores to tokens representing the correct information. This observation persists across various question types and knowledge domains. The pattern is particularly evident in specialized domains, where models might respond with "unsure" while assigning high probabilities to correct technical terms. This suggests that traditional evaluation methods focusing solely on the model's final output may substantially underestimate the knowledge actually stored in the model's parameters.

## 2.3 THE HITS@$k$ METRIC

Building on these observations, we propose the Hits@$k$ metric to quantify the model's knowledge retention:

$$\text{Hits@}k = \frac{N^k_{correct}}{N} \tag{1}$$

where $N^k_{correct}$ represents cases where the correct answer appears within the top-k logits. For large vocabulary models such as LLAMA3 with approximately 128,000 tokens, we find that a relatively small k value effectively captures stored knowledge while maintaining computational efficiency.

Experimental results in Figure 4 demonstrate the effectiveness of this metric in revealing stored knowledge. Using LLAMA3-8B on DBpedia, while Hits@1 is only 17.2%, Hits@5 reaches 57.9%, indicating substantially more stored knowledge than suggested by traditional metrics. This pattern holds across different domains and data types, suggesting a fundamental characteristic of how LLMs store and access information. These findings motivate a deeper examination of factors affecting knowledge storage and expression, which we explore in Section 3.

## 3 EVALUATING SETUP

### 3.1 DATASETS

To evaluate our approach, we conduct experiments on both open-domain and domain-specific datasets. DBPedia represents an open-domain dataset, encompassing general knowledge across various fields. For domain-specific evaluation, IMDB contains movie-related information while GoodReads focuses on book-related knowledge.

Following Sun et al. (2023), we partition the data into head, torso, and tail portions based on entity frequency, with head containing the top 10% most frequent entities. This dataset selection enables analysis of both domain characteristics and popularity effects on memory patterns.

### 3.2 MODELS AND IMPLEMENTATION

We conduct experiments using the following LLMs: LLAMA2-13B, LLAMA2-70B, LLAMA3-8B, LLAMA3-70B, LLAMA3.1-8B, QWEN2-1.5B, QWEN2-7B, QWEN2-72B, MISTRAL-7B-INSTRUCT-V0.3 (abbreviated as MISTRAL-7B). These models represent different architectural choices and parameter scales, ranging from 1.5B to 70B parameters. To minimize randomness in model outputs, we use greedy decoding with temperature set to 0.0 across all experiments.

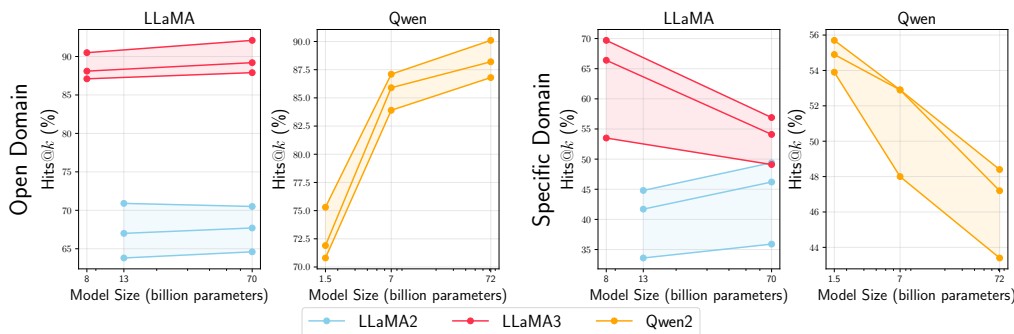

Figure 2: The Hits@$k$ scores of different large language models on the DBPedia-Head dataset when $k = 100$.

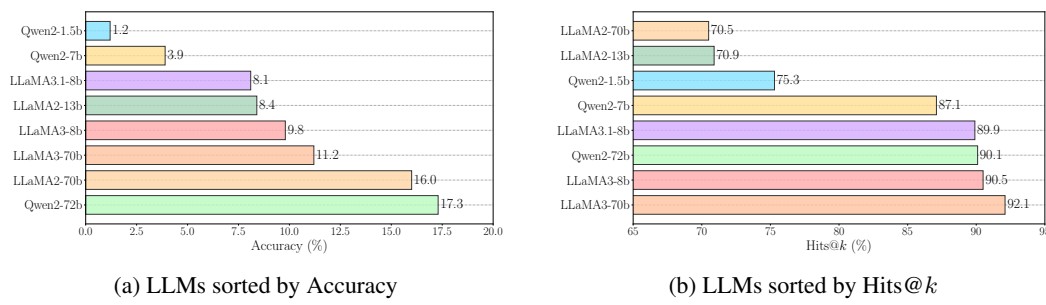

(a) LLMs sorted by Accuracy  (b) LLMs sorted by Hits@$k$

Figure 3: The ranking of LLMs based on Accuracy and Hits@$k$ on DBPedia-Head when $k = 100$.

### 3.3 EVALUATION PROTOCOL

Given that some models utilize subword tokenization, we employ string comparison to assess whether the model's output matches the correct answer. Specifically, if any of the top-$k$ tokens share at least three consecutive characters with the ground truth, we classify that token as a match. The value of k in Hits@$k$ correlates with the model's vocabulary size, particularly important for larger models like LLAMA3 with approximately 128,000 tokens. For questions where the model lacks confidence, we allow it to respond with "unsure" as outlined in our prompt design.

## 4 ANALYSIS AND RESULTS

### 4.1 OVERALL PERFORMANCE

#### 4.1.1 HITS@$k$ PERFORMANCE OF DIFFERENT MODELS

**A larger model size does not mean a higher Hits@$k$ score**  Figure 2 shows the results on dataset DBPedia-head, demonstrating this finding. As the number of parameters increases, LLMs exhibit improved accuracy across a range of tasks, including QA tasks. This is due to the greater representational power of larger models, allowing them to capture more nuanced and complex language patterns. However, among the three datasets used for testing, the Hits@$k$ results for the LLAMA2-13B and LLAMA2-70B models are similar, as are the Hits@$k$ results for the LLAMA3-8B and LLAMA3-70B models. As shown in Figure 3, the rankings of LLMs based on Accuracy and Hits@$k$ differ significantly. This indicates that increasing the model size does not necessarily lead to richer or more comprehensive memory in LLMs.

**Newer LLMs have higher Hits@$k$ scores**  Our experimental results indicate that newer LLMs exhibit higher Hits@$k$. For instance, the Hits@$k$ of LLaMA3 significantly surpasses that of LLaMA2, regardless of model size. In the head section of the DBPedia dataset, the LLAMA3-70B model achieves a score of 92.1%, the LLAMA3-8B model scores 90.5%, and the LLAMA2-70B model

scores 70.5%. This suggests that newer models have a more comprehensive memory of relevant knowledge, likely due to updates in training data. In particular, newer datasets tend to encompass a broader range of information.

**Justification for Hits@k as a Knowledge Metric.** We posit that Hits@k captures genuine latent knowledge rather than surface-level token co-occurrence. This validity is supported by three key observations: (1) the systematic nature of the storage-expression gap across diverse domains Figure 2; (2) the distinct model rankings yielded by Hits@k compared to standard accuracy, indicating it measures a fundamental internal property Figure 3, which empirically proves these tokens represent accessible, usable knowledge suppressed by decoding dynamics.

### 4.1.2 ANALYSIS OF THE INFLUENCE OF $k$ VALUE SELECTION

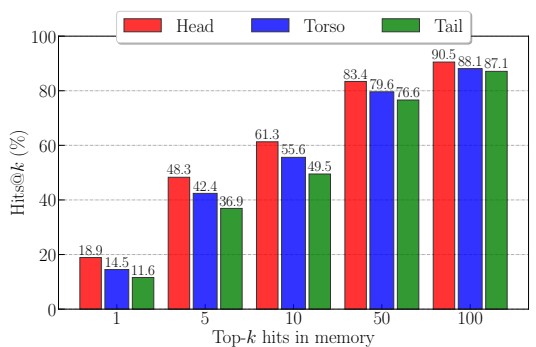

Figure 4 presents the Hits@$k$ scores for various $k$ values. A lower Hits@1 score suggests that the model struggles to provide the correct answer directly in the QA task. However, as the $k$ value increases, the score improves, indicating that the model retains relevant knowledge. Our experimental results indicate that, when $k$ = 50, the Hits@$k$ for the head, torso, and tail sections exceeds 80%. Despite the extensive vocabulary of the LLaMA3 model (approximately 128,000 tokens), the correct answer is frequently located within a relatively small number of tokens at the begin-

Figure 4: For different values of $k$, We report the Hits@$k$ of LLAMA3-8B on the DBpedia dataset.

ning. This suggests that the model has the potential to provide correct answers in most cases, even if an incorrect answer is initially generated. We observe a significant difference between scores at $k$ = 1 and $k$ = 5, indicating that utilizing tokens with higher probabilities can yield more reliable answers.

We show the cumulative distribution of the ranks of Hits@$k$ score in the QA task in Figure 5. We observe that the difference in popularity has a smaller impact on Hits@$k$ for the DBPedia dataset compared to IMDB. This suggests that the domain of datasets influences the sensitivity to popularity. Generally, memory performance on open-domain datasets is less sensitive to variations in popularity.

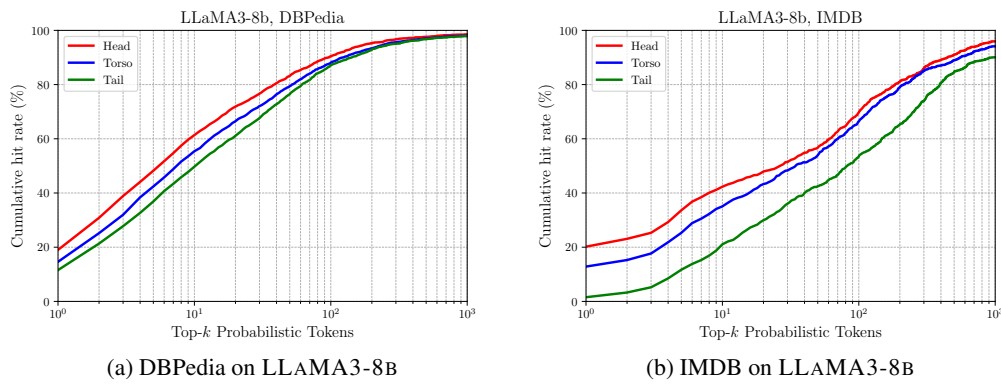

(a) DBPedia on LLAMA3-8B          (b) IMDB on LLAMA3-8B

Figure 5: The cumulative distribution of the ranks of Hits@$k$ in the QA task

## 4.2 CROSS-DOMAIN ANALYSIS

### 4.2.1 COMPARISON OF OPEN DOMAIN AND SPECIFIC DOMAIN

As shown in Table 1, The Hits@$k$ results of data sets in different domains are different. DBLP is an open-domain dataset, while IMDB and Goodreads are domain-specific datasets. The experimental results demonstrate that the Hits@$k$ for the open-domain dataset is higher than that for the domain-specific datasets.

We show the cumulative distribution of the ranks of Hits@$k$ score in the QA task in Figure 5. We observe that the difference in popularity has a smaller impact on Hits@$k$ for the DBPedia dataset compared to IMDB. This suggests that the domain of the dataset influences the sensitivity to popularity. Generally, memory performance on open-domain datasets is less sensitive to variations in popularity.

Table 1: Experimental results (Hits@$k$, $k$ = 100) for models of varying sizes were obtained by testing different popularity subsets of the head-to-tail dataset.

| $k$ = 100 | DBPedia | | | IMDB | | | GoodReads | | |
|---|---|---|---|---|---|---|---|---|---|
| | Head | Torso | Tail | Head | Torso | Tail | Head | Torso | Tail |
| **LLAMA2-13B** | 70.9 | 67.0 | 63.8 | 44.8 | 41.7 | 33.6 | 36.5 | 36.5 | 28.6 |
| **LLAMA2-70B** | 70.5 | 67.7 | 64.6 | 49.4 | 46.2 | 35.9 | 36.1 | 35.6 | 31.0 |
| **LLAMA3-8B** | 90.5 | 88.1 | 87.1 | 69.7 | 66.4 | 53.5 | 67.8 | 68.5 | 65.6 |
| **LLAMA3-70B** | 92.1 | 89.2 | 87.9 | 56.9 | 54.1 | 49.1 | 44.2 | 45.4 | 43.0 |
| **LLAMA3.1-8B** | 89.9 | 87.5 | 86.0 | 69.3 | 67.0 | 53.0 | 67.8 | 68.3 | 65.3 |
| **QWEN2-1.5B** | 75.3 | 71.9 | 70.8 | 53.9 | 48.0 | 43.4 | 37.8 | 38.1 | 35.3 |
| **QWEN2-7B** | 87.1 | 85.9 | 83.9 | 54.9 | 52.9 | 47.2 | 41.9 | 43.0 | 41.7 |
| **QWEN2-72B** | 90.1 | 88.2 | 86.8 | 55.7 | 52.9 | 48.4 | 43.8 | 44.8 | 41.7 |
| **MISTRAL-7B** | 73.8 | 69.8 | 66.2 | 50.6 | 45.7 | 35.7 | 35.3 | 34.8 | 29.4 |

### 4.2.2 THE IMPACT OF DOMAIN ON KNOWLEDGE STORAGE

**Specific domain datasets are more susceptible to memory loss** Our experimental results show that, compared to open-domain datasets, the Hits@$k$ of specific-domain datasets is lower, indicating that LLMs are more prone to memory loss in specific-domain datasets. This phenomenon may be due to the fact that certain knowledge in specific-domain datasets is not included in the model's training data.

## 4.3 POPULARITY IMPACT

**Popularity impacts the model's memory storage, though to a lesser extent.** Our experiments indicate that the popularity of datasets influences Hits@$k$. Within the same domain, higher popularity correlates with higher Hits@$k$. However, the difference in Hits@$k$ is smaller than the difference observed when directly calculating the model's accuracy in QA tasks. This suggests that, beyond the training data, the degree of memory expression significantly impacts the model's accuracy across datasets with varying popularity. Specifically, in datasets with lower popularity, the model is more likely to retain knowledge related to the questions but may still fail to provide the correct answers.

**Popularity exerts a greater influence in specific-domain datasets** Our experiments demonstrate that popularity has a greater impact on Hits@$k$ in specific domain datasets. This suggests that, compared to open-domain datasets, popularity significantly influences memory storage in specific-domain datasets, making it more likely for the model to lack relevant memory in less popular specific-domain datasets.

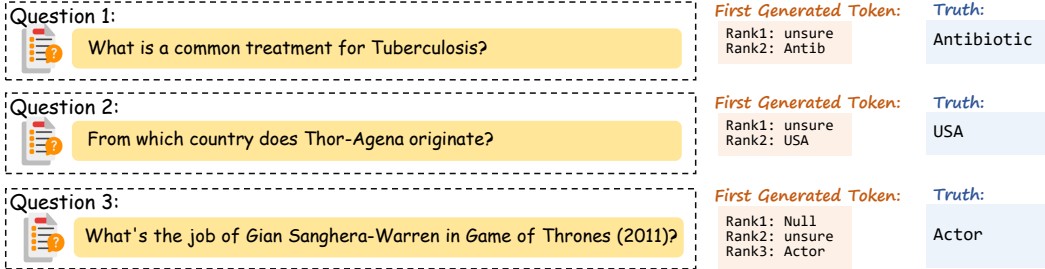

Figure 6: Case study when the LLAMA3-8B model produces uncertain answers. In Question 1 and Question 2, the model's final answer is "unsure", but the correct answer, or a subword related to it, appears in the second position of the logits. In Question 3, the model's final answer is a blank character, which is deemed incorrect. However, the correct answer appears in the token with the third-highest probability.

### 4.4 UNINFORMATIVE RESPONSE IMPACT

A noteworthy phenomenon is that, in some cases, the model's response is uninformative. This includes instances of: 1) repeating specific strings, and 2) outputting empty strings, among others. These types of responses are labeled as "error". Such responses may arise from anomalies in the model's generation process, or from a lack of relevant memory.

To reduce the likelihood of the model providing incorrect answers, prompts in QA tasks often include an "unsure" option, allowing the model to respond with "unsure" when uncertain about the correct answer. This approach helps minimize the risk of hallucinations when the model encounters unfamiliar information. Figure 7 shows the distribution ratios of three response types under the LLAMA3-8B model: uninformative, correct, and wrong. We found that uninformative responses have a greater impact on the model's performance. Our experiments revealed that when some models answered "unsure", they still retained relevant knowledge in memory. This suggests that the model may respond with "unsure" even when relevant memory exists. We show an example of this situation in Figure 6.

In summary, we classify the cases mentioned above as uninformative responses, with Figure 6 showing the proportion of such responses across different datasets. We show two different types of uninformative responses in Figure 6, the correct answers appear in the tokens corresponding to the second-highest or third-highest logits. In Questions 1 and 2, the model's final answer is "unsure," yet the correct answer, or a related subword, appears as the second most probable token in the logits. In Question 3, the model's final answer is a null character, which is considered incorrect. However, the correct answer is found in the token with the third-highest probability. This indicates that, while the model possesses relevant memory, it fails to output the correct answer.

As shown in Figure 7, in the DBPedia dataset, experimental results show that more than half of the responses in the Head, Torso, and Tail sub-datasets are uninformative. In the domain-specific IMDB dataset, the high proportion of uninformative responses also significantly impacts the model's accuracy in QA tasks. This highlights the significant impact of uninformative responses on the final results in both open-domain and domain-specific datasets. Moreover, as dataset popularity decreases, the proportion of uninformative responses increases, which emerges as a key factor contributing to the decline in accuracy in QA tasks.

Our experiments, however, indicate that even uninformative responses may still contain relevant knowledge memory. Fully automating the identification and filtering of incorrect answers is challenging, but identifying and filtering uninformative responses is comparatively straightforward. Since identifying these responses is straightforward, filtering them and extracting the model's latent knowledge for QA tasks can effectively improve the model's performance.

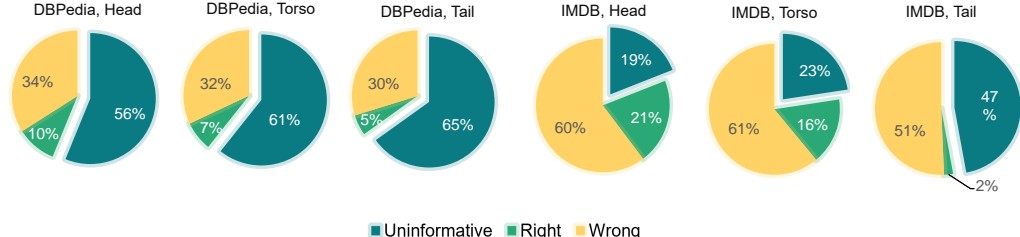

Figure 7: We present the distribution of three response types: uninformative, right, and wrong. Additionally, we analyzed data from both open domain and specific domain datasets, reporting the experimental results for the LLAMA3-8B model.

Table 2: Answer recovery rates from "unsure" responses on DBPedia (left) and IMDB (right) datasets. Filtering uninformative tokens reveals a substantial portion of correct answers masked during initial decoding.

| DBPedia | Greedy decoding | | | Decoding with Unsure filter | | |
|---|---|---|---|---|---|---|
| | Head | Torso | Tail | Head | Torso | Tail |
| LLAMA2-13B | 8.4 | 4.0 | 3.2 | 13.5 ↑5.1 | 8.9 ↑4.9 | 6.9 ↑3.7 |
| LLAMA2-70B | 16.0 | 12.3 | 8.1 | 18.3 ↑2.3 | 14.0 ↑1.7 | 9.4 ↑1.3 |
| LLAMA3-8B | 9.8 | 7.3 | 5.1 | 13.6 ↑3.8 | 10.5 ↑3.2 | 7.6 ↑2.5 |
| LLAMA3-70B | 11.2 | 8.7 | 6.0 | 23.0 ↑11.8 | 18.1 ↑9.4 | 12.7 ↑6.7 |
| LLAMA3.1-8B | 8.1 | 5.3 | 3.7 | 15.6 ↑7.5 | 10.1 ↑4.8 | 7.4 ↑3.7 |
| QWEN2-1.5B | 1.2 | 0.7 | 0.6 | 2.4 ↑1.2 | 1.5 ↑0.8 | 1.4 ↑0.8 |
| QWEN2-7B | 3.9 | 2.5 | 1.3 | 9.3 ↑5.4 | 7.5 ↑5.0 | 4.8 ↑3.5 |
| QWEN2-72B | 17.3 | 12.1 | 9.0 | 20.1 ↑2.8 | 14.3 ↑2.2 | 10.2 ↑1.2 |
| MISTRAL-7B | 16.5 | 11.0 | 7.5 | 16.7 ↑0.2 | 11.3 ↑0.3 | 7.9 ↑0.4 |

| IMDB | Greedy decoding | | | Decoding with Unsure filter | | |
|---|---|---|---|---|---|---|
| | Head | Torso | Tail | Head | Torso | Tail |
| LLAMA2-13B | 15.7 | 11.1 | 0.0 | 21.9 ↑6.2 | 17.6 ↑6.5 | 2.0 ↑2.0 |
| LLAMA2-70B | 25.2 | 23.5 | 4.3 | 25.2 | 23.5 | 4.5 ↑0.2 |
| LLAMA3-8B | 20.7 | 16.4 | 2.2 | 21.3 ↑0.6 | 17.1 ↑0.7 | 2.7 ↑0.5 |
| LLAMA3-70B | 19.1 | 18.6 | 4.3 | 25.4 ↑6.3 | 24.0 ↑5.4 | 4.9 ↑0.6 |
| LLAMA3.1-8B | 18.3 | 13.4 | 2.0 | 18.8 ↑0.5 | 14.5 ↑1.1 | 2.3 ↑0.3 |
| QWEN2-1.5B | 2.0 | 0.9 | 0.5 | 2.5 ↑0.5 | 1.1 ↑0.2 | 0.5 |
| QWEN2-7B | 7.4 | 2.9 | 0.3 | 11.7 ↑4.3 | 4.5 ↑1.6 | 0.4 ↑0.1 |
| QWEN2-72B | 19.1 | 16.2 | 1.1 | 20.3 ↑2.2 | 18.1 ↑1.9 | 1.2 ↑0.1 |
| MISTRAL-7B | 20.5 | 15.1 | 1.3 | 20.5 | 15.2 ↑0.1 | 1.4 ↑0.1 |

# 5 REVISITING "UNSURE" RESPONSES IN KNOWLEDGE-BASED QA

## 5.1 OBSERVATION

In many knowledge-based question answering (KBQA) benchmarks, models are permitted to respond with "unsure" when they lack confidence in producing a correct answer. This strategy is commonly employed to reduce the risk of hallucinations or factually incorrect outputs. As illustrated in Figure 7, a non-negligible portion of model predictions fall into this uninformative category.

However, this raises an important question: when a model generates "unsure", does it truly lack the relevant knowledge, or is the correct answer being suppressed due to low decoding confidence? To investigate this, we analyze the token-level logit distributions during generation. Surprisingly, we observe that in a significant number of "unsure" cases, the correct answer still appears among the top-$k$ (e.g., $k = 2$ or $3$) candidates by logit rank, even though the model ultimately selects the "unsure" token as the output.

## 5.2 QUANTITATIVE ANALYSIS

The observations above suggest that LLMs may retain correct knowledge even when they abstain from answering. To systematically measure this phenomenon, we design a set of controlled experiments to quantify the extent to which correct answers are recoverable from "unsure" outputs.

Our goal is to evaluate the latent presence of correct answers in the model's internal distributions, independent of what is ultimately generated. Specifically, we analyze the top-$k$ tokens ranked by logit scores in cases where the model initially outputs "unsure." We then apply a filtering procedure to remove uninformative candidates (e.g., "unsure", null strings, or stop words), and identify whether the remaining tokens contain the correct answer.

This procedure allows us to estimate the gap between knowledge storage and expression, revealing how often the model possesses the correct information but fails to surface it due to confidence calibration or decoding dynamics. The details of this method are described below.

To quantify this phenomenon, we propose a simple two-stage decoding procedure that filters out "unsure"-related tokens and re-invokes the model for answer generation. The decoding pipeline is outlined in Algorithm 1.

Given a question $q$, let $P(t \mid q)$ denote the model's probability distribution over the vocabulary $V$, and let $T_k = \{t_1, \ldots, t_k\}$ be the top-$k$ tokens ranked by logit scores. We define a token $t$ as *uninformative* if it satisfies any of the following heuristics: it begins with "uns", corresponds to an empty string, contains fewer than three characters, or consists solely of stop words. Let $U$ denote the set of such uninformative tokens. We select the highest-probability informative token from $T_k$ as the candidate answer $a^*$:

$$a^* = \arg \max_{t \in T_k \setminus U} P(t \mid q) \tag{2}$$

Token $a^*$ is then appended to the original prompt and fed back into the model to trigger a new round of decoding.

Using this method, we find that a significant fraction of previously "unsure" responses can be successfully recovered as correct answers. This suggests that the model often retains latent knowledge internally, but refrains from surfacing it due to conservative decoding or over-cautious uncertainty thresholds. We emphasize that the "Unsure" Filtering Decoding strategy is designed strictly as an *analytical probe* to quantify the memory-masking effect, rather than as a deployment-ready method.

Table 2 report the recovery rates observed on two QA benchmarks after applying our decoding strategy. These results highlight the potential of knowledge recovery mechanisms in enhancing factual completeness without compromising model reliability.

## 6 RELATED WORK

**Question-Answering tasks and Hallucination for LLMs** Question-Answering (QA) tasks have become a central application area for LLMs. A key challenge in their adoption and optimization is addressing hallucination, where LLMs generate incorrect or unsupported informationHuang et al. (2023). Currently, there are numerous benchmarks available for evaluating QA tasks on LLMs Berant et al. (2013); Joshi et al. (2017); Dubey et al. (2019); Kwiatkowski et al. (2019); Sciavolino et al. (2021); Mallen et al. (2022); Kumar et al. (2024); Zhong

---

**Algorithm 1** Decoding Without "Unsure" Tokens

**Require:** Token list $L$ (ranked by logit scores), original prompt $\text{Prompt}_{\text{old}}$
1: $i \leftarrow 0$
2: **while** $L[i]$ is uninformative **do**
3:      Remove $L[i]$ from $L$
4:      $i \leftarrow i + 1$
5: **end while**
6: $a^* \leftarrow L[i]$
7: $\text{Prompt}_{\text{new}} \leftarrow \text{Prompt}_{\text{old}} + a^*$
8: $\text{Output}_{\text{new}} \leftarrow \text{LLM}(\text{Prompt}_{\text{new}})$

---

et al. (2023). Sun et al. (2023) proposed datasets partitioned based on popularity. Tonmoy et al. (2024) analyzed the challenges and limitations for hallucination mitigation. Zhang et al. (2023b) analyzed various types of hallucinations in LLMs. Gu et al. (2022) proposed a generic framework and trained a discriminator to evaluate probability of candidate plans for QA tasks. Du et al. (2023) used correlation analysis techniques to quantify and locate the sources of hallucinations, aiming to enhance the reliability of the model. Zhu et al. (2024) evaluated the model's performance on hallucination problems in real-world scenarios, especially on knowledge-intensive question answering tasks. Waldo & Boussard (2024) analyzed the root causes of hallucinations in large language models and discussed possible directions for improvement.

**LLMs as Knowledge Bases** Previous work has proposed that pre-trained language models can be used as knowledge bases Petroni et al. (2019); AlKhamissi et al. (2022). Petroni et al. (2019)

introduced the LAMA benchmark, which consists of questions formatted as "fill-in-the-blank" cloze statements. He et al. (2024) explores the potential of LLMs in memorizing exact knowledge in large-scale knowledge bases. Zhong et al. (2023) pointed out the multi-hop knowledge editing problem when using LLMs as knowledge bases. Zheng et al. (2024) investigates the potential of LLMs as knowledge bases, especially in knowledge-intensive tasks. Singhal et al. (2023) demonstrates the potential of LLMs in encoding medical knowledge and answering medical questions. Pan et al. (2024) presents a forward-looking roadmap for the unification of LLMs and Knowledge Graphs (KGs). Yin et al. (2024) offers a fresh perspective and a new method for evaluating large language models, which can help in more accurately understanding and assessing the performance of these models. Hu et al. (2023) evaluates the factual knowledge of LLMs using a benchmark called Pinocchio, which includes 20,000 diverse questions. It finds that while LLMs can implicitly store facts, they often lack accuracy and are unable to update or reason over multiple facts effectively.

## 7 CONCLUSION

We investigate how large language models (LLMs) express stored knowledge in question answering (QA) tasks. To this end, we propose metrics for assessing latent knowledge retention beyond surface-level correctness. Analysis across datasets of varying popularity and domain shows that memory expression correlates with dataset familiarity. Notably, correct answers often appear among top-ranked tokens even when the model outputs "unsure". By filtering out uninformative tokens, we reveal a gap between knowledge storage and expression, offering insights for improved prompting and decoding in knowledge-intensive tasks.

## ETHICS STATEMENT

This work does not involve human subjects, sensitive data, animal experiments, or any other aspect that raises ethical concerns. No potential risks of misuse or negative societal impact have been identified.

## REPRODUCIBILITY STATEMENT

We are committed to ensuring reproducibility of our results. All code, along with instructions for data preprocessing, model configuration, and evaluation, will be released upon publication to enable full replication of the experiments and results reported in this paper.

## ACKNOWLEDGEMENTS

Jing Tang's work is partially supported by the National Natural Science Foundation of China (NSFC) under Grant No. 62402410 and by Guangdong Provincial Project (No. 2023QN10X025).

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

## A  DATASET DETAILS

To ensure reproducibility and alignment with established benchmarks, we strictly adhered to the dataset settings and partition strategies provided by Sun et al. (2023). We utilized three datasets covering both open-domain and specific-domain knowledge: DBPedia, IMDB, and GoodReads. The specific configurations for each dataset are as follows:

- **DBPedia (Open Domain):** This dataset serves as our open-domain knowledge source and is a knowledge graph derived from Wikipedia. Consistent with the configuration in Sun et al. (2023), we utilized the English snapshot from December 1, 2022.
- **IMDB (Specific Domain - Movie):** For the movie domain, we utilized the IMDB dataset. Following the settings in Sun et al. (2023), we used the data snapshot from May 21, 2023.
- **GoodReads (Specific Domain - Book):** For the book domain, we employed the GoodReads dataset. This utilizes the 2017 crawl data originally published by Wan & McAuley (2018) and subsequently adopted by Sun et al. (2023).

**Data Partitioning.**  Following the methodology of Sun et al. (2023), each dataset is partitioned into three subsets based on entity popularity: *Head*, *Torso*, and *Tail*. The *Head* partition contains the top 10% most frequent entities, allowing for a granular analysis of the model's knowledge retention across varying degrees of entity frequency.

## B  USE OF LARGE LANGUAGE MODELS

Large language models (LLMs) were used solely for language polishing and minor editorial assistance (e.g., grammar, wording, and clarity). They were not involved in the conception of research ideas, design of experiments, data analysis, or interpretation of results. All scientific content, methods, and conclusions were developed independently by the authors.

## C  DETAILS OF PROMPTS

The few-shot prompt used in this paper for the QA task is shown below, with two examples provided to guide the LLM in generating the correct answer.

> Answer the following questions in as few words as possible. Say "unsure" if you don't know.
> Question: What is the capital of China?
> Answer: Beijing
> Question: What is the captical of Wernythedia?
> Answer: unsure
> Question: [question]
> Answer:

## D  MORE EXPERIMENTAL DATA

As shown in Tables 3 to 5, we show the Hits@$k$ performance of different types of models when $k$ takes different values.

Table 3: Experimental results (Hits@$k$, $k$ = 5) for models of varying sizes were obtained by testing different popularity subsets of the head-to-tail dataset.

| $k$ = 5 | DBPedia | | | IMDB | | | GoodReads | | |
|---|---|---|---|---|---|---|---|---|---|
| | Head | Torso | Tail | Head | Torso | Tail | Head | Torso | Tail |
| **LLAMA2-13B** | 23.0 | 17.2 | 15.1 | 15.4 | 10.0 | 2.5 | 15.1 | 12.5 | 2.4 |
| **LLAMA2-70B** | 25.9 | 19.6 | 17.4 | 17.8 | 13.6 | 3.7 | 16.6 | 14.9 | 6.4 |
| **LLAMA3-8B** | 48.3 | 42.4 | 36.9 | 33.6 | 25.2 | 11.5 | 30.5 | 28.7 | 16.3 |
| **LLAMA3-70B** | 57.8 | 50.0 | 43.1 | 34.7 | 30.4 | 10.9 | 27.5 | 27.8 | 16.9 |
| **LLAMA3.1-8B** | 48.0 | 41.2 | 36.0 | 30.9 | 23.2 | 10.7 | 45.6 | 40.3 | 30.7 |
| **QWEN2-1.5B** | 23.3 | 19.6 | 17.1 | 13.6 | 7.0 | 3.8 | 7.3 | 5.5 | 2.9 |
| **QWEN2-7B** | 46.1 | 40.6 | 35.2 | 35.7 | 24.5 | 18.2 | 22.5 | 14.7 | 8.4 |
| **QWEN2-72B** | 54.5 | 45.4 | 38.7 | 34.3 | 26.2 | 5.2 | 25.9 | 21.9 | 12.2 |
| **MISTRAL-7B** | 32.8 | 25.0 | 20.6 | 23.1 | 15.6 | 5.5 | 16.7 | 12.0 | 3.5 |

Table 4: Experimental results (Hits@$k$, $k$ = 10) for models of varying sizes were obtained by testing different popularity subsets of the head-to-tail dataset.

| $k$ = 10 | DBPedia | | | IMDB | | | GoodReads | | |
|---|---|---|---|---|---|---|---|---|---|
| | Head | Torso | Tail | Head | Torso | Tail | Head | Torso | Tail |
| **LLAMA2-13B** | 31.5 | 25.1 | 21.0 | 20.2 | 13.3 | 3.5 | 19.0 | 16.1 | 4.3 |
| **LLAMA2-70B** | 34.9 | 28.4 | 24.1 | 25.2 | 20.6 | 6.9 | 20.9 | 18.9 | 8.0 |
| **LLAMA3-8B** | 61.3 | 55.6 | 49.5 | 42.5 | 35.3 | 20.7 | 52.4 | 48.8 | 39.2 |
| **LLAMA3-70B** | 67.9 | 60.3 | 54.3 | 41.8 | 35.7 | 19.7 | 31.3 | 30.1 | 20.1 |
| **LLAMA3.1-8B** | 61.4 | 55.1 | 49.3 | 39.6 | 31.4 | 23.1 | 53.8 | 48.8 | 38.7 |
| **QWEN2-1.5B** | 32.9 | 29.5 | 24.5 | 22.4 | 16.2 | 8.6 | 11.1 | 8.2 | 6.6 |
| **QWEN2-7B** | 55.8 | 50.7 | 46.2 | 41.7 | 32.5 | 24.3 | 26.3 | 19.8 | 13.6 |
| **QWEN2-72B** | 62.4 | 55.2 | 50.4 | 40.2 | 34.2 | 22.3 | 29.8 | 26.1 | 16.9 |
| **MISTRAL-7B** | 42.1 | 34.4 | 28.4 | 28.1 | 19.8 | 8.6 | 21.0 | 15.0 | 5.5 |

Table 5: Experimental results (Hits@$k$, $k$ = 50) for models of varying sizes were obtained by testing different popularity subsets of the head-to-tail dataset.

| $k$ = 50 | DBPedia | | | IMDB | | | GoodReads | | |
|---|---|---|---|---|---|---|---|---|---|
| | Head | Torso | Tail | Head | Torso | Tail | Head | Torso | Tail |
| **LLAMA2-13B** | 57.7 | 52.1 | 47.2 | 38.1 | 32.3 | 19.8 | 29.5 | 27.7 | 18.2 |
| **LLAMA2-70B** | 59.5 | 54.3 | 49.1 | 44.9 | 39.0 | 26.4 | 30.0 | 28.9 | 22.0 |
| **LLAMA3-8B** | 83.4 | 79.6 | 76.6 | 69.7 | 53.8 | 42.2 | 63.1 | 63.0 | 58.6 |
| **LLAMA3-70B** | 86.7 | 82.5 | 79.0 | 55.0 | 49.5 | 41.5 | 39.2 | 40.8 | 35.4 |
| **LLAMA3.1-8B** | 82.4 | 78.8 | 76.0 | 58.1 | 53.9 | 42.6 | 63.3 | 61.6 | 57.4 |
| **QWEN2-1.5B** | 59.8 | 54.7 | 52.4 | 47.3 | 40.9 | 31.3 | 27.3 | 23.3 | 22.6 |
| **QWEN2-7B** | 78.7 | 74.5 | 73.0 | 52.6 | 49.1 | 41.3 | 37.7 | 36.3 | 32.0 |
| **QWEN2-72B** | 83.6 | 79.5 | 77.5 | 53.9 | 48.6 | 41.4 | 39.1 | 38.0 | 35.5 |
| **MISTRAL-7B** | 63.2 | 57.5 | 52.7 | 44.9 | 38.4 | 25.7 | 29.2 | 26.7 | 18.6 |

