# OpenReview forum: "Are LLMs Really Not Knowledgeable? Mining the Submerged Knowledge in LLMs' Memory"
_ICLR.cc/2026/Conference — ICLR 2026 Poster_

### Official Review · Reviewer_Jtaf · 2025-10-20

**Soundness:** 3
**Presentation:** 3
**Contribution:** 2
**Rating:** 2
**Confidence:** 4

**Summary:**

This paper investigates whether large language models truly “do not know” an answer when they produce an incorrect response. The authors argue that model knowledge and model expression are not equivalent, and they conduct exploratory experiments such as Hits@k and “unsure” filtering to show that correct answers often appear in the top-k token distribution, even when surface accuracy is low. The work suggests that LLMs may possess latent knowledge that is not successfully expressed during decoding.

**Strengths:**

- Clear motivation. The paper highlights an intuitively important gap between latent knowledge and surface-level generation in LLMs.

- Empirical observations are easy to interpret. Hits@k and “unsure” filtering provide simple and intuitive diagnostic signals.

- Readable paper structure. The writing is clear, and the experiments are straightforward to follow.

**Weaknesses:**

- The insight is not novel, as similar conclusions have long existed in perplexity-based evaluations, which already reflect that LLMs may assign high probability to correct tokens that are not selected in top-1 decoding.

- The phenomenon is also well-known from rollout-based methods (e.g., multi-sampling, self-consistency, and RL trajectories), which routinely reveal correct answers in non-greedy decoding paths.

- The paper lacks deeper analysis or actionable contribution, offering no explanation of why the mismatch occurs, nor methods for leveraging latent knowledge to improve actual model performance.

**Questions:**

See above.

---

> ### Author Response · Authors · 2025-11-21
>
> We sincerely thank the reviewer for their rigorous critique and high standards regarding the novelty and depth of our work. We appreciate the opportunity to clarify the specific positioning of our contributions relative to existing evaluation paradigms. We believe that while the general existence of latent knowledge is known, our work provides a necessary systematic quantification and a specific mechanism analysis (the storage-expression gap) that distinguishes it from broad metrics like perplexity or sampling heuristics. Below, we provide a detailed response.
>
> **Weakness 1:** The paper introduces Hits@k as a metric to explicitly reveal and quantify the latent knowledge stored in LLMs. However, the reviewer suggests that this insight may not be novel, as perplexity-based evaluations already capture the probability assignments of tokens, implying that the phenomenon of high-probability correct tokens is already implicitly understood in the field.
>
> **R 1:** We appreciate the reviewer’s extensive knowledge of evaluation metrics. We would like to clarify the distinction between the aggregate nature of perplexity and the specific, retrieval-oriented insight provided by our framework.
>
> While perplexity measures the model's general predictive uncertainty (average log-likelihood), it does not explicitly quantify factual retrieval success. A model might have low perplexity (predicting common words well) but still fail to retrieve a specific entity.  The paper proposes Hits@k to specifically evaluate latent knowledge retention independent of answer surface form, effectively isolating the factual knowledge capability by focusing solely on the rank of the correct entity token.
>
> The novelty lies not just in the existence of probability, but in the magnitude of the systematic gap between storage and expression that we uncover. For instance, we show that on DBpedia, LLAMA3-8B has a standard accuracy of 17.2% but a Hits@5 of 57.9%.   This massive disparity (over 40%) highlights a structural misalignment in how models utilize their parameters—a specific diagnostic insight that standard perplexity scores do not reveal.
>
> By explicitly separating Hits@k (storage) from Accuracy (expression), we provide a tool to diagnose why a model fails.  Our work proves that the failure often stems from "expression issues rather than knowledge gaps", which fundamentally alters the direction of optimization (focusing on decoding/alignment rather than just pre-training).
>
>
> **Weakness 2:** The paper identifies that correct answers are often maintained in the model's output distributions even when not selected by greedy decoding. However, the reviewer notes that this phenomenon is also observed in rollout-based methods (such as multi-sampling, self-consistency, and RL trajectories) and questions whether our findings offer new insights beyond what these existing methods already utilize.
>
> **R 2:** We thank the reviewer for pointing out the connection to sampling-based methods. We view our work as providing the theoretical explanation and mechanism for why those methods work, rather than competing with them as a decoding strategy.
>
> Methods like self-consistency rely on sampling but do not explain why the correct answer appears in stochastic paths despite being rejected by greedy decoding. Our analysis of the logit distributions reveals the mechanism: the correct answer is often strictly ordered as the second or third ranked token. We provide the empirical evidence that this "submerged" knowledge is highly accessible and structured, not just randomly scattered in the tail.
>
> Beyond general non-greedy decoding, we specifically identify a novel memory-masking effect triggered by uncertainty. We show that prompting strategies allowing "unsure" responses can actively suppress correct answers that are otherwise highly ranked. This specific interaction between prompt design (safety/uncertainty) and logit suppression is a key insight that general rollout studies have not systematically quantified.

---

> > ### Author Response · Authors · 2025-11-21
> >
> > **Weakness 3**: The paper conducts an analysis of the mismatch between knowledge storage and expression. However, the reviewer feels that the current version lacks deeper analysis into the causes of this mismatch or actionable contributions on how to leverage this latent knowledge to actually improve model performance.
> >
> > **R 3**: We appreciate the reviewer’s push for deeper analysis and actionable outcomes. We effectively agree that understanding the root causes is critical, and we would like to clarify the specific contributions of our work in this regard.
> >
> >  We conducted a detailed analysis of the factors influencing this mismatch. Our experiments reveal that data popularity and domain specificity are key causal factors: popularity impacts knowledge storage (Hits@k) less than it impacts expression (Accuracy) . This provides the deep insight that models often store tail knowledge well, but the "expression bottleneck" blocks it more severely than head knowledge.
> >
> >  Regarding actionable contributions, our goal was to scrutinize the prevailing few-shot QA paradigm rather than propose a new deployment algorithm . We identified that prompting strategies which permit "unsure" responses—a common practice to reduce hallucination—can inadvertently suppress correct answers by discouraging low-confidence generation.
> >
> > Consequently, the "Unsure" Filtering Decoding strategy (Algorithm 1) is designed as an analytical probe to quantify this "memory-masking effect" . By successfully recovering a subset of masked answers, we provide empirical proof that the current QA paradigm often hides usable knowledge .  This offers actionable guidance for future prompt and decoding design, suggesting that reliance on simple "unsure" prompts may be overly conservative.
> >
> > We once again thank the reviewer for their rigorous critique, which has pushed us to sharpen the positioning of our work relative to existing evaluation paradigms. We hope that our detailed responses regarding the distinction between Hits@k and perplexity, the mechanistic explanation we provide for non-greedy decoding phenomena, and the analytical purpose of our uninformative token filtering have adequately addressed your concerns about novelty and contribution. We respectfully request that the reviewer considers these clarifications and reassesses the value of our paper.

---

> > > ### Comment · Reviewer_Jtaf · 2025-11-27
> > >
> > > Thank you very much for your careful work and detailed responses. I have updated my score accordingly

---

> > > > ### Author Response · Authors · 2025-11-28
> > > >
> > > > We are sincerely grateful for your positive reassessment and for raising the score. We deeply appreciate the time and effort you dedicated to reviewing our work. Your constructive feedback has been instrumental in clarifying our contributions and improving the overall quality of our manuscript.

---

### Official Review · Reviewer_JvA3 · 2025-10-23

**Soundness:** 3
**Presentation:** 3
**Contribution:** 2
**Rating:** 4
**Confidence:** 4

**Summary:**

The paper shows that LLMs can retain correct knowledge even when generating incorrect answers; correct answers frequently appear among high-probability tokens despite not being selected as final outputs. Based on this observation, the paper introduces Hits@k, a new metric to assess the knowledge of LLMs. Also, it introduces a new decoding method to improve answer accuracy by leveraging detected but unexpressed knowledge.

**Strengths:**

- This paper catches an interesting finding that LLMs often maintain access to accurate information within their probability distributions over vocabulary tokens, and there is a systematic gap between knowledge storage and expression rather than simple knowledge absence.

- It offers new insights into knowledge augmentation: instead of expanding knowledge, augmenting the ability to express existing knowledge is important and can be potentially very useful.

**Weaknesses:**

- Though it is an interesting finding, I still believe that LLMs are not knowledgeable even though they assign significant probability scores to tokens representing the correct information, since in real-world use cases, it is impractical to let LLMs generate multiple responses to each query. Therefore, I don't think Hits@k should be used for evaluation/rank models.

- The proposed decoding algorithm can raise many safety or ethical concerns if deployed into general use cases, since in many real-world scenarios, it might be unsafe or unethical to generate an “informative” response.

- The proposed decoding algorithm increases the probability of correct answers, but also increases the probability of wrong answers.

**Questions:**

Cite and introduce DBPedia, IMDB, and GoodReads with more details (maybe in appendix).

---

> ### Author Response · Authors · 2025-11-21
>
> We sincerely thank the reviewer for their thoughtful evaluation and for finding our discovery regarding the gap between probability scores and final outputs interesting. We appreciate the valid concerns raised regarding the practical utility of the Hits@k metric and the safety implications of our decoding strategy. These perspectives are valuable for refining the scope and application of our work. Below, we address these points in detail.
>
>
> **Weakness 1:** The paper presents Hits@k as a measure of latent knowledge. However, the reviewer argues that since real-world applications typically require a single correct response, Hits@k may not be suitable for ranking models, as possessing high-probability correct tokens does not necessarily make an LLM "knowledgeable" in a practical sense.
>
> **R 1:** We appreciate the reviewer’s practical perspective on model evaluation. We would like to clarify that the primary goal of our work is to diagnose the source of errors—specifically, distinguishing between "not knowing" (storage failure) and "not saying" (expression failure)—rather than proposing Hits@k as a sole metric for deployment readiness.
>
> * Diagnostic Value: As mentioned in the Introduction of our submission, prior work often attributes QA failures to knowledge gaps in the model's parameters, leading to solutions focused on expanding model capacity or training data. Our work challenges this assumption by using Hits@k to prove that the knowledge is often present but suppressed. This is a crucial scientific distinction: it suggests that future improvements should focus on alignment and decoding (unlocking expression) rather than just scaling up pre-training (increasing storage).
> * Latent vs. Surface Knowledge: We agree that for an end-user, only the final output matters. However, for model developers, Hits@k reveals the "submerged" potential of the model. As shown in Figure 3, the ranking of models changes significantly between Accuracy and Hits@k. This indicates that some models are much more knowledgeable in their parameters than their surface performance suggests, which is a valuable insight for model selection and fine-tuning.
> * Conclusion: We will follow the reviewer's suggestion to clarify in the paper that Hits@k is intended as a diagnostic metric for latent capacity, while standard accuracy remains the metric for deployment performance.
>
> **Weakness 2:** The paper introduces a decoding algorithm to filter "unsure" tokens. However, the reviewer raises important safety concerns, noting that in general use cases, forcing an "informative" response could be unsafe or unethical.
>
> **R 2:** We thank the reviewer for raising this important ethical consideration. We effectively agree that safety is paramount in general-purpose deployment.
>
> * Context of Factual QA: Our experiments are conducted on standard factual benchmark datasets (DBPedia, IMDB, GoodReads). In this specific context, "unsure" responses typically stem from low confidence in the factual answer rather than safety refusals (e.g., harmful content).
> * Proof-of-Concept Nature: The decoding algorithm in Algorithm 1 is designed primarily as an analytical probe to quantify the "memory-masking effect". It demonstrates that the model can answer correctly if forced, proving the knowledge exists.
>
> **Weakness 3:** The paper aims to improve answer recovery. However, the reviewer notes that while the proposed algorithm increases the probability of correct answers, it inevitably also increases the probability of incorrect answers.
>
> **R 3:** We appreciate the reviewer’s accurate observation of this trade-off. We would like to clarify that the decoding algorithm in Algorithm 1 is primarily designed as an analytical probe to quantify the gap between knowledge storage and expression, rather than a deployment strategy to enhance model performance in real-world applications. Our goal is not to propose a new decoding standard for general use, but to use this method experimentally to reveal how often the model possesses the correct information but fails to surface it due to confidence calibration.
>
> Our objective is to scrutinize the prevailing few-shot QA paradigm and quantitatively demonstrate that prompting strategies permitting "unsure" outputs can inadvertently suppress correct answers. This allows us to measure the "memory-masking effect" where latent knowledge is hidden behind conservative generation.
>
>  We added a clarification that this method trades precision (by removing the "unsure" option) for recall (recovering latent knowledge), which is appropriate for measuring model capacity but requires careful tuning for user-facing applications.

---

> > ### Author Response · Authors · 2025-11-21
> >
> > **Question 1:** The paper relies on DBPedia, IMDB, and GoodReads for evaluation. The reviewer requests that these datasets be cited and introduced with more detail, specifically suggesting the addition of an appendix.
> >
> > **R 1:** We thank the reviewer for this helpful suggestion to improve the reproducibility and clarity of our work.
> >
> > * **Adoption of Established Benchmarks:** We would like to clarify that we did not construct these datasets ourselves. Instead, we strictly adhered to the dataset settings and partition strategies provided by [1] to ensure alignment with established benchmarks.
> > * **Specific Dataset Details:** Following the reviewer's suggestion, we have added detailed descriptions of the data sources in the Appendix. Specifically, consistent with the configuration in [1]: The DBPedia dataset (Open Domain) is a knowledge graph based on Wikipedia from [1], utilizing the English snapshot from December 1, 2022. The IMDB dataset (Movie) utilizes the snapshot from May 21, 2023. The Goodreads dataset (Book) utilizes the 2017 crawl data originally published by [2] and adopted by [1].
> >
> > We have updated the Appendix to include these specific details, construction methods, and citations to ensure full transparency.
> >
> >
> > We once again thank the reviewer for their constructive feedback, which has significantly helped us improve the clarity and rigor of our work. We hope that our detailed responses regarding the diagnostic nature of Hits@k, the safety considerations of our decoding strategy, and the dataset specifications have adequately addressed your concerns.
> >
> > **References:**
> > [1] Sun, K., Xu, Y., Zha, H., Liu, Y., & Dong, X. L. Head-to-tail: How knowledgeable are large language models (llms)? aka will llms replace knowledge graphs?.\
> > [2] Wan, M., & McAuley, J. Item recommendation on monotonic behavior chains. In Proceedings of the 12th ACM conference on recommender systems .

---

### Official Review · Reviewer_h4kU · 2025-11-01

**Soundness:** 3
**Presentation:** 3
**Contribution:** 3
**Rating:** 6
**Confidence:** 3

**Summary:**

The paper investigates how large language models (LLMs) store and express factual knowledge. It argues that incorrect or "unsure" answers do not necessarily indicate missing knowledge, since correct answers often appear among high-probability tokens that are not selected. To quantify this hidden knowledge, the authors introduce Hits@k, which measures how frequently the correct answer appears within the top-k tokens of the model's output distribution. Extensive experiments across open-domain and domain-specific datasets show that models retain substantially more factual information than is revealed by accuracy alone, and that newer models exhibit higher latent retention. The study also finds that "unsure" prompts can suppress correct answers by lowering generation confidence, and that filtering such responses can recover many correct predictions. Together, these findings reveal a gap between knowledge storage and expression, offering insights for improving prompt design and decoding strategies in knowledge-intensive tasks.

**Strengths:**

- The paper introduces a clear and intuitive metric that captures latent knowledge beyond standard accuracy, offering a new perspective on model evaluation.

- The analysis reveals that models often “know” more than they express, which challenges common assumptions about what low-confidence or incorrect outputs imply.

- The experiments are extensive and well controlled, which show consistent trends across multiple model scales and factual datasets.

- The study provides actionable insights for prompt design and decoding strategies by showing how uncertainty affects knowledge expression.

- The paper is clearly written and conceptually accessible, making its findings easy to reproduce and useful for both research and applied settings.

**Weaknesses:**

- The paper does not provide a formal justification for why Hits@k should reflect internal knowledge rather than distributional coincidence, relying mainly on empirical correlations (Figure 3).

- The improvement margins between Hits@k and standard accuracy are sometimes modest -- for example, less than 5% in several datasets (Table 2) -- which weakens the claim of large hidden knowledge reserves.

- The evaluation focuses narrowly on factual recall and omits reasoning or multi-hop questions, so it is unclear whether the proposed metric captures deeper forms of knowledge use beyond surface recall (Section 5.2).

- The proposed method measures the presence of correct tokens but ignores how easily the model can retrieve or reason about them, which conflates memorization with accessibility (Section 4.3).

- The study does not examine sensitivity to decoding parameters such as temperature or top-p, leaving unclear whether the observed patterns remain stable under different generation settings.

**Questions:**

How does Hits@k distinguish between genuinely stored knowledge and coincidental token co-occurrence, and what evidence supports that the correct token's presence in the top-k reflects meaningful internal representation rather than surface-level probability alignment?

---

> ### Author Response · Authors · 2025-11-21
>
> We sincerely thank the reviewers for their time and constructive feedback. We are encouraged by the reviewers' recognition of our work's novelty in uncovering the gap between knowledge storage and expression in Large Language Models. We appreciate the helpful suggestions regarding the justification of our metric and the scope of our evaluation, which have helped us clarify our contributions and improve the quality of our manuscript. Below, we provide detailed responses to the specific concerns.
>
> **Weakness 1:** The paper successfully uncovers a significant phenomenon—the gap between knowledge storage and expression in LLMs. However, the reviewer questions whether the proposed Hits@k metric truly reflects genuinely stored internal knowledge or merely represents surface-level token co-occurrence, suggesting that the empirical correlation might not be sufficient for formal justification.
>
> **R 1:** We thank the reviewer for their critical and insightful question regarding the theoretical underpinning of our key metric, Hits@k. We would like to clarify the reviewer’s misunderstanding on the systematic nature of our observation, which supports Hits@k as a measure of latent knowledge, as follows.
>
> * Our argument is built upon the observation of a systematic gap between knowledge storage and expression. This gap is not a coincidence but is visually exemplified in Figure 1, where the correct answer, "Olympia," retains a high probability score even when an incorrect output is selected. This pattern persists across various knowledge domains and model architectures, suggesting a systematic issue with knowledge expression rather than random noise.
> * Furthermore, the fundamental difference in the ranking of LLMs based on standard Accuracy (Hits@1) versus Hits@k shown in Figure 3 demonstrates that Hits@k captures a distinct property of the model's internal state beyond the surface output.
> * Most compellingly, our Section 5 analysis shows that the correct answer is frequently present among the top-k tokens (e.g., k=2 or k=3) even when the model actively abstains from answering by outputting "unsure". We then showed a set of experiments where this knowledge can be successfully recovered by filtering the uninformative tokens. This demonstrated recoverability strongly indicates that the presence of the correct token in the top-k logits reflects latent, usable knowledge, not just random token co-occurrence.
> * We thank the reviewer for highlighting the need for clearer justification of the metric's validity. We follow the reviewer’s suggestion to add a more detailed discussion on the theoretical basis of Hits@k in the revised manuscript to improve our paper.
>
> **Weakness 2:** The paper claims that LLMs possess significantly more factual knowledge than conventional metrics suggest. However, the reviewer notes that the improvement margins shown in Table 2 (Answer Recovery Rate) are sometimes modest, and wonders if this weakens the paper's claim of large hidden knowledge reserves.
>
> **R 2:** We appreciate the reviewer’s careful scrutiny of our numerical results. We would like to clarify the reviewer’s misunderstanding on the experimental context of Table 2.
>
> * Table 2 reports the Answer Recovery Rate from responses that initially output "unsure". This experiment is designed to quantify the memory-masking effect of over-cautious generation on a specific subset of questions where the model abstains, not the total latent knowledge.
> * The primary evidence supporting our claim of large hidden knowledge reserves is found in the direct comparison between Hits@k and standard Accuracy (Hits@1). For example, in Section 2.3, the LLAMA3-8B model achieves only 17.2% standard accuracy (Hits@1) but reaches 57.9% for Hits@5 on DBpedia. This is an improvement of over 40 percentage points.
> * Furthermore, our comprehensive results show that on the DBPedia-Head dataset, the LLAMA3-8B model reaches a Hits@k (k=100) score of 90.5%, while its standard accuracy is only 9.8%. This massive difference firmly substantiates our claim of substantial knowledge storage being under-expressed.
> * We thank the reviewer for allowing us to clarify the interpretation of the results in Table 2 versus the fundamental Hits@k findings. We follow the reviewer's suggestion to improve the clarity of our paper by emphasizing this distinction.

---

> > ### Author Response · Authors · 2025-11-21
> >
> > **Weakness 3:** The paper provides a robust analysis of the knowledge storage-expression gap in knowledge-intensive tasks. However, the reviewer points out that the evaluation is narrowly focused on single-hop factual recall and suggests that the proposed metric's ability to capture deeper forms of knowledge use, such as reasoning or multi-hop knowledge, remains unclear.
> >
> > **R 3:** We thank the reviewer for their insightful comment on the scope of our evaluation. We would like to clarify the rationale behind our strategic focus on single-hop factual recall.
> >
> >
> > Foundational Importance of Factual QA: Our research investigates the capabilities of LLMs as "parametric knowledge bases". In this context, the ability to accurately access atomic factual information is the most fundamental requirement. Single-hop QA serves as the bedrock for complex reasoning; if a model cannot correctly express stored atomic facts due to the "storage-expression gap", it
> > inevitably fail in multi-hop tasks that depend on these facts. Therefore, ensuring the reliable expression of single-hop knowledge is a prerequisite for more advanced applications.
> >
> > Our primary goal is to rigorously identify and quantify the "systematic gap between knowledge storage and expression". Single-hop QA provides the cleanest experimental setting to isolate this phenomenon from the confounding factors of reasoning logic or context maintenance. This allows us to conclusively demonstrate that model failures often stem from "expression issues rather than knowledge gaps", thereby establishing the storage-expression gap as a distinct and significant problem that must be addressed at the fundamental level.
> >
> > **Weakness 4:** The paper introduces Hits@k to quantify latent knowledge retention. However, the reviewer accurately notes the risk of conflating knowledge memorization (storage) with knowledge accessibility and reasoning (expression) when using a metric like Hits@k.
> >
> > **R 4:** We appreciate the reviewer’s insightful analysis that highlights the distinction between knowledge storage and knowledge expression. We would like to clarify the reviewer’s misunderstanding on how our framework addresses this very distinction.
> >
> > * The fundamental purpose of Hits@k is precisely to quantify knowledge storage (memorization) independent of surface output. The difference between the high Hits@k score and the low Accuracy (Hits@1) score is the core metric we use to quantify the storage-expression gap.
> > * We further explore this accessibility concern in Section 4.3 through the impact of popularity. We find that the influence of popularity on Hits@k (storage) is smaller than its influence on standard Accuracy (expression). This demonstrates that for less popular data, the model is more likely to retain knowledge (high Hits@k) but less likely to express it correctly (low Accuracy).
> > * Thus, our method actively utilizes the contrast between Hits@k and Accuracy to explore the balance between memorization and accessibility. We thank the reviewer for this prompt and will strengthen the language in Section 4.3 to emphasize how our metric dissects these two concepts.
> >
> > **Weakness 5:** The paper provides a robust analysis of LLMs' memory under greedy decoding. However, the reviewer suggests that the observed patterns might be influenced by the fixed generation settings (greedy decoding with T=0.0), making it unclear if they generalize to different decoding strategies (e.g., non-zero temperature or top-p sampling).
> >
> > **R 5:** We thank the reviewer for proposing a valuable area for further testing and validation.
> >
> > In this foundational study, we deliberately employed greedy decoding with a temperature set to 0.0 across all experiments. This choice was made to eliminate randomness and ensure that the Hits@k measure reflects the knowledge encoded directly in the model's logits (internal memory). We agree with the reviewer that performing an analysis of temperature or top-p sensitivity is an important next step. Given our empirical observation that the correct answer is often present just one or two ranks below the top-1 prediction (e.g., Rank 2 or 3 in the "unsure" cases), we hypothesize that minor adjustments in these parameters would likely serve as an effective mechanism to surface more of this latent knowledge.

---

> > > ### Author Response · Authors · 2025-11-21
> > >
> > > **Question 1:** The paper proposes Hits@k as a metric to measure latent knowledge. However, the reviewer asks for stronger evidence to show how Hits@k distinguishes between genuinely stored knowledge and coincidental token co-occurrence, and what supports that the correct token's presence in the top-k reflects meaningful internal representation rather than surface-level probability alignment.
> > >
> > > **R 1:** We thank the reviewer for this insightful question, which reinforces the core justification of our proposed metric. The evidence supporting the top-k presence of the correct token as meaningful internal representation is multifaceted:
> > >
> > > * Systematic and Large Disparity: We observe a systematic and significant gap between the highest-ranked token (Hits@1) and the presence of the correct token within the top-k (Hits@k). For LLAMA3-8B on DBpedia, Hits@1 is 17.2%, while Hits@5 is 57.9%. This large, consistent disparity across models and domains suggests a genuine knowledge accessibility bottleneck rather than a random token co-occurrence.
> > > * Successful Recovery from Abstentions: Our strongest evidence is the successful recovery of answers from "unsure" outputs. Case studies (Figure 6) show the correct answer token often sitting at Rank 2 or 3 when the model chooses to output "unsure" (Rank 1). When we filter out the "unsure" token, we successfully recover a significant fraction of these correct answers (Table 2). This ability to retrieve the information proves that the high-ranked token represents usable, stored knowledge.
> > > * Systematic Variation with Domain: The Hits@k scores systematically decrease when moving from the open-domain DBPedia dataset to the more specific IMDB/GoodReads datasets (Table 1). This drop is precisely what is expected for a metric measuring stored knowledge, as specific domain knowledge is less likely to be present in general pre-training data.
> > >
> > > We thank the reviewer for encouraging us to clearly summarize the empirical foundation of our metric.

---

### Meta-Review · Area_Chair_D8yx · 2026-01-06

**Summary:**

This paper investigates the gap between knowledge storage and expression in LLMs, introducing Hits@k to quantify latent factual knowledge beyond standard accuracy. It shows that correct answers frequently reside in the top-k tokens even when the model outputs an incorrect or "unsure" response.

The work presents a clear, well-articulated observation for the community. While the core phenomenon is not entirely new, the paper provides a focused quantification and a specific analysis of the "unsure"-induced suppression effect. The primary concern is the magnitude of the claimed contribution versus perceived incrementalism.

**Reviewer Concerns:**

Reviewer 1: The rebuttal has addressed most specific concerns.

Reviewer 2: The authors' responses adequately address the raised concerns.

Reviewer 3: This reviewer's concern regarding the novelty of the core insight remains the most significant point of discussion.

**Reviewer Scores:**

Reviewer #2 and #3 might slightly increase their score.

---

### Decision · Program_Chairs · 2026-01-26

Accept (Poster)